# What went wrong and when?
# Instance-wise feature importance for time-series black-box models

**Sana Tonekaboni**[*]
Department of Computer Science
University of Toronto, Vector institute
stonekaboni@cs.toronto.edu

**Shalmali Joshi**[*][†]
Harvard University (SEAS)
shalmali@seas.harvard.edu

**Kieran R Campbell**
Department of Computer Science
University of Toronto, Vector institute
Lunenfeld Tanenbaum Research Institute
kieran.campbell@utoronto.ca

**David Duvenaud**
Department of Computer Science
University of Toronto, Vector institute
duvenaud@cs.toronto.edu

**Anna Goldenberg**
Department of Computer Science
University of Toronto, Vector institute
The Hospital for Sick Children
anna.goldenberg@utoronto.ca

## Abstract

Explanations of time series models are useful for high stakes applications like healthcare but have received little attention in machine learning literature. We propose FIT, a framework that evaluates the importance of observations for a multivariate time-series black-box model by quantifying the shift in the predictive distribution over time. FIT defines the importance of an observation based on its contribution to the distributional shift under a KL-divergence that contrasts the predictive distribution against a counterfactual where the rest of the features are unobserved. We also demonstrate the need to control for time-dependent distribution shifts. We compare with state-of-the-art baselines on simulated and real-world clinical data and demonstrate that our approach is superior in identifying important time points and observations throughout the time series.

## 1 Introduction

Understanding what drives machine learning models to output a particular prediction can aid reliable decision-making for end-users and is critical in high stakes applications like healthcare. This problem has been particularly overlooked in the context of time series datasets compared to static settings. Time series domain is unique because the data's dynamic nature results in the features driving model prediction changing over time. Explaining time-series machine learning model prediction can be defined in many ways, with existing works primarily considering two approaches. The first class uses the notions of instance-level feature importance proposed in the literature for static supervised

---

[*]Equal author
[†]Work done while at the Vector Institute

learning [36, 35, 7, 21]. These either compute the gradients of the output with respect to the input vector or perturb features to evaluate their impact on model output. None of the approaches of this type explicitly model the temporal dependencies that exist in time series data. The second class uses attention models explicitly designed for time series. A number of works show that attention scores can be interpreted as an importance score for different observations over time within the context of these models [8, 34, 14].

In this work, we propose Feature Importance in Time (FIT), a framework to quantify the importance of observations over time, based on their contribution to the temporal shift of the model output distribution. Our proposed score quantifies how well an observation approximates the predictive shift at every time-step, and our score improves over existing notions of instance-level feature importance over time. The proposed method is model-agnostic and can thus be applied to any black-box model in a time series setting. Our contributions are as follows:

1. We pose the problem of assigning importance to observations of a time series as that of quantifying the contribution to the predictive distributional shift under a KL-divergence by contrasting the predictive distribution against a counterfactual where the remaining features are unobserved.

2. We use generative models to learn the dynamics of the time series. This allows us to model the distribution of measurements, and therefore accurately approximate the counterfactual effect of subsets of observations over time.

3. Unlike existing approaches, FIT allows us to evaluate the aggregate importance of subsets of features over time, which is critical in healthcare contexts where simultaneous changes in feature subsets drive model predictions [2].

## 2 Preliminaries

Let $\mathbf{X} \in \mathbb{R}^{D \times T}$ be a sample of a multi-variate time-series data where $D$ is the number of features with $T$ observations over time. Further, $\mathbf{x}_t \in \mathbb{R}^D$ is the set of all observations at time $t \in 1, \ldots, T$, denoted by the vector $[x_{1,t}, x_{2,t}, \cdots, x_{D,t}]$ and $\mathbf{X}_{0:t} \in \mathbb{R}^{D \times t}$ is the matrix $[\mathbf{x}_0; \mathbf{x}_1; \cdots; \mathbf{x}_t]$. Let $\mathbf{S} \subseteq \{1, 2, \cdots, D\}$ be some subset of the features with corresponding observations at time $t$ denoted by $\mathbf{x}_{\mathbf{S},t}$. Similarly, the set $\mathbf{S}^c = \{1, 2, \cdots, D\} \setminus \mathbf{S}$ indicates the complement set of $\mathbf{S}$ with corresponding observations $\mathbf{x}_{\mathbf{S}^c,t}$. For a single feature, $\mathbf{S} = \{d\}$, an observation at time $t$ is indicated by $x_{d,t}$ and the rest of the observations are denoted by $\mathbf{x}_{-d,t}$. Additional details on the notation used for exposition work is summarized in the Appendix A.1. We are interested in explaining a black-box model $f_\theta$ that estimates the conditional distribution $p(y_t|\mathbf{X}_{0:t})$ at every time step $t$, using observations up to that time point, $\mathbf{X}_{0:t} \in \mathbb{R}^{D \times t}$.

## 3 FIT: Feature Importance in Time

Our goal is to assign an importance score $I(\mathbf{x}_\mathbf{S}, t)$ to a set of observations $\mathbf{x}_{\mathbf{S},t}$ corresponding to a subset of features $\mathbf{S}$ at time $t$. An important observation is one that best approximates the outcome distribution, even in the absence of other features. We formalize this by evaluating whether the partial conditional distribution $p(y|\mathbf{X}_{0:t-1}, \mathbf{x}_{\mathbf{S},t})$ for a set of observations closely approximates the full predictive distribution $p(y|\mathbf{X}_{0:t})$ at time $t$. Note that there can be an underlying temporal shift in the predictive distribution due to additional data being acquired over time. Therefore the importance of a subset of observations $\mathbf{x}_{\mathbf{S},t}$ should be evaluated relative to this shift. We characterize the distributional shift from the previous time point $t-1$ to time instance $t$ by the KL-divergence of the associated distributions i.e. $\mathrm{KL}(p(y|\mathbf{X}_{0:t} \parallel p(y|\mathbf{X}_{0:t-1}))$. When a new observation $\mathbf{x}_{\mathbf{S},t}$ is obtained at time $t$, its relative importance to other observations (or subset of observations) can be evaluated in terms of whether it *adds* additional information that may explain the change in predictive distribution from the previous time-step. We formalize this intuition using the following importance assignment score:

**FIT importance Score:** We define $I(\mathbf{x}_\mathbf{S}, t)$ to be the importance of new set of observations $\mathbf{S}$ at time $t$, which is quantified by the shift in predictive distribution when only $\mathbf{S}$ are observed while accounting for the distributional shift over time:

$$I(\mathbf{x}_\mathbf{S}, t) = \underbrace{\mathrm{KL}(p(y|\mathbf{X}_{0:t}) \parallel p(y|\mathbf{X}_{0:t-1}))}_{\text{T1: temporal distribution shift}} - \underbrace{\mathrm{KL}(p(y|\mathbf{X}_{0:t}) \parallel p(y|\mathbf{X}_{0:t-1}, \mathbf{x}_{\mathbf{S},t}))}_{\text{T2: unexplained distribution shift}} \tag{1}$$

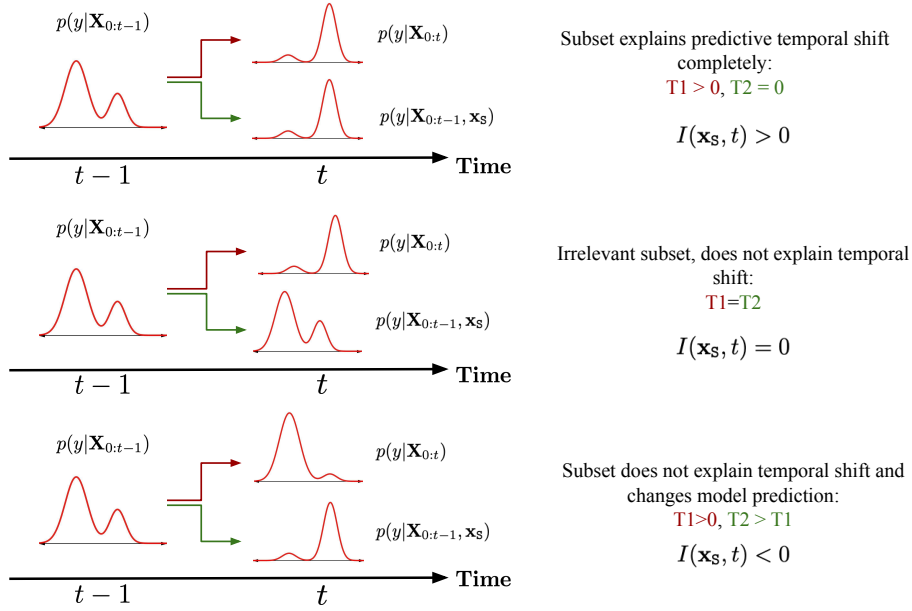

Figure 1: Pictorial depiction of FIT score assignment. Each panel shows possible cases that reflect different regimes of the FIT score. Top panel: Positive temporal shift that is completely explained by observations $\mathbf{x_S}$ i.e. $\mathbf{x_S}$ is an important subset of observations. Middle panel: Observing subset $\mathbf{x_S}$ only does not explain temporal shift. Bottom panel: Subset $\mathbf{x_S}$ does not explain temporal shift and in fact, changes the estimate of the predictive distribution.

As shown in equation (1), the score is composed of two terms. The first term estimates the distributional shift between the predictive distribution at time $t-1$ to $t$. The second KL term measures the residual shift after adding only $\mathbf{x}_{S,t}$. Note that $\mathbf{X}_{0:t} = [\mathbf{X}_{0:t-1}; [\mathbf{x}_{S,t}, \mathbf{x}_{S^c,t}]]$. Thus, the second KL-term measures the residual shift when only $\mathbf{x}_{S,t}$ is added and $\mathbf{x}_{S^c,t}$ is unobserved. Therefore, the overall score measures how important observing $\mathbf{x}_{S,t}$ would be for estimating the output distribution at time $t$. Also, the direction of the KL-divergence is chosen such that the difference between the two distributions is weighted by the original predictive distribution.

The score can also be written in terms of the cross-entropy $\mathrm{H}(P, Q)$ between the associated distributions:

$$I(\mathbf{x_S}, t) = \mathrm{H}\left(p(y|\mathbf{X}_{0:t}), p(y|\mathbf{X}_{0:t-1})\right) - \mathrm{H}\left(p(y|\mathbf{X}_{0:t}), p(y|\mathbf{X}_{0:t-1}, \mathbf{x}_{S,t})\right) \tag{2}$$

and so can be interpreted as measuring the average number of additional bits of information $\mathbf{x}_{S,t}$ provides at time $t$ in approximating the full distribution.

Our proposed score has several desirable properties. Firstly, consider all subsets with cardinality $|\mathbf{S}| = 1$: as $D \to \infty$, we have T1 = T2 and any one feature is unable to approximate the predictive distribution meaning $I(\mathbf{x_S}, t) = 0$. Secondly, when $|S| = D$ (i.e. we wish to ascertain how important *all* features are), then T2 = 0 and the importance score is maximal.

To gain further intuition into the behavior of the proposed score, consider the different scenarios as depicted in Figure 1. A maximal positive importance score (Figure 1, top) indicates that if only $\mathbf{x}_{S,t}$ were observed at time $t$, the full predictive distribution $p(y|\mathbf{X}_{0:t})$ is captured and thus S is *important*. In contrast, a score of 0 (Figure 1, middle) indicates that the outcome distribution has changed over time but S no longer reflects the predictive distribution at time $t$. Finally, a negative score (Figure 1, bottom) indicates that adding S alone in fact worsens the approximation of $p(y|\mathbf{X}_{0:t})$.

For reliable estimation of the importance score, we need an accurate approximation of the partial predictive distribution $p(y|\mathbf{X}_{0:t-1}, \mathbf{x}_{S,t})$. The approximation can be written as an expectation term, described in Equation (3). We use Monte-Carlo integration to estimate this expectation by sampling

---

**Algorithm 1 FIT**
**Input:** $f_\theta$: Trained Black-box predictor model, time series $\mathbf{X}_{0:T}$, where $T$ is the max time, L: **Number of Monte-Carlo samples**, S: **A subset of features of interest.**
**Output: Importance score matrix** I

---

1: Train $\mathcal{G}$ using $\mathbf{X}_{0:T}$
2: **for all** $t \in [T]$ **do**
3:　　$p(y|\mathbf{X}_{0:t}) = f_\theta(\mathbf{X}_{0:t})$
4:　　$p(y|\mathbf{X}_{0:t-1}) = f_\theta(\mathbf{X}_{0:t-1})$
5:　　$p(\mathbf{x}_t|\mathbf{X}_{0:t-1}) \approx \mathcal{G}(\mathbf{X}_{0:t-1})$
6:　　**for all** $l \in [\mathtt{L}]$ **do**
7:　　　　Sample $\hat{\mathbf{x}}_{\mathtt{S}^c,t}^{(l)} \sim p(\mathbf{x}_{\mathtt{S}^c,t}|\mathbf{X}_{0:t-1}, \mathbf{x}_{\mathtt{S},t})$
8:　　　　$p(\hat{y}^{(l)}) = f_\theta(\mathbf{X}_{0:t-1}, \mathbf{x}_{\mathtt{S},t}, \hat{\mathbf{x}}_{\mathtt{S}^c,t}^{(l)})$
9:　　$p(y|\mathbf{X}_{0:t-1}, \mathbf{x}_{\mathtt{S},t}) \approx \frac{1}{L} \sum_{l=1}^{L} p(\hat{y}^{(l)})$
10:　　$I(\mathbf{x}_{\mathtt{S}}, t) = \text{KL}(p(y|\mathbf{X}_{0:t}) \parallel p(y|\mathbf{X}_{0:t-1})) - \text{KL}(p(y|X_{0:t}) \parallel p(y|X_{0:t-1}, \mathbf{x}_{\mathtt{S},t}))$
11: Return $I(\mathbf{x}_{\mathtt{S}}, t)$

---

unobserved counterfactuals from the distribution conditioned on subset S. $p(y|\mathbf{X}_{0:t-1}, \mathbf{x}_{\mathtt{S},t}, \hat{\mathbf{x}}_{\mathtt{S}^c,t})$ in Equation 3 is approximated by the black-box model $f_\theta$.

$$p(y|\mathbf{X}_{0:t-1}, \mathbf{x}_{\mathtt{S},t}) = \mathbb{E}_{\hat{\mathbf{x}}_{\mathtt{S}^c,t} \sim p(\mathbf{x}_{\mathtt{S}^c,t}|\mathbf{X}_{0:t-1}, \mathbf{x}_{\mathtt{S},t})}[p(y|\mathbf{X}_{0:t-1}, \mathbf{x}_{\mathtt{S},t}, \hat{\mathbf{x}}_{\mathtt{S}^c,t})] \tag{3}$$

FIT uses a generative model $\mathcal{G}$ to estimate the distribution of measurements conditioned on the past, $p(\mathbf{x}_t|\mathbf{X}_{0:t-1})$ to approximate the counterfactual distribution $p(\mathbf{x}_{\mathtt{S}^c,t}|\mathbf{X}_{0:t-1}, \mathbf{x}_{\mathtt{S},t})$. We use a recurrent latent variable generator, introduced in Chung et al. [9] to model $p(\mathbf{x}_t|\mathbf{X}_{0:t-1})$ using a multivariate Gaussian with full covariance matrix that encodes potential correlation between features. Conditioning this distribution on the observation of interest, provides a realistic conditional distribution for the counterfactual observations. Note that the specifics of the generative model architecture can be a design choice. We have chosen a recurrent generator since it allows us to model non-stationarity in the time series while handling varying length of observations.

## 3.1 Feature importance assignment algorithm

The proposed procedure is summarized in Algorithm 1. We assume that we are given a trained black-box model $f_\theta$, that estimates the predictive distribution $p(y|\mathbf{X}_{0:t})$ for every $t$, and the data it had been trained on (labels are not required). We first train a generator $\mathcal{G}$ to learn the conditional distribution $p(\mathbf{x}_t|\mathbf{X}_{0:t-1})$. For a specific subset of interest S, the counterfactual distribution is obtained by conditioning $p(\mathbf{x}_{\mathtt{S}^c,t}|\mathbf{X}_{0:t-1})$ on S i.e. $p(\mathbf{x}_{\mathtt{S}^c,t}|\mathbf{X}_{0:t-1}, \mathbf{x}_{\mathtt{S},t})$. At every time point, we approximate the partial output distribution using Monte-Carlo samples drawn from this conditional distribution. We then assign score based on the formula presented in Equation (1). For ease of interpretation, the score $I(\mathbf{x}_{\mathtt{S}}, t)$ may be normalized.

## 3.2 Subset importance

One of the key characteristics of FIT is its ability to evaluate the joint importance of subsets of features. This is a highly desirable property in cases where simultaneous changes in multiple measurements drive an outcome. For example in a healthcare setting, uncorrelated changes in heart rate and blood pressure can be indicative of a clinical instability while the converse is not necessarily concerning. This property of FIT can be used in 3 different scenarios:

1. For evaluating instance-level feature importance when $|\mathtt{S}| = 1$. This task is comparable to what most feature importance algorithms do.

2. In cases where there are pre-defined set of features, for instance a set of blood works in the clinic, FIT can be used to find the aggregate importance of those features together (see Appendix B.7 for an example).

3. For finding the minimal set of observations to acquire in a time series setting, to approximate the predictive distribution well.

# 4    Experiments

We evaluate our feature importance assignment method (FIT) on a number of simulated data (where ground-truth feature importance is available) and more complex real-world clinical tasks. We compare with multiple groups of baseline methods, described below:

1. Perturbation-based methods: Two approaches are used as a baseline in this category. In **Feature Occlusion (FO)** [32], importance is assigned based on the difference in model prediction when each feature $x_i$ is replaced with a random sample from the uniform distribution. As an augmented alternative, we introduce **Augmented Feature Occlusion (AFO)**, where we replace observations with samples from the bootstrapped distribution $p(x_i)$ for each feature $i$. This is to avoid generating out-of-distribution noise samples.

2. Gradient-based methods: This includes methods that decompose the output on input features by backpropagating the contribution to every feature of the input. We have included (a) **Deep-Lift** [26] and (b) **Integrated gradient (IG)** [31] for this comparison.

3. Attention-based method **(RETAIN):** This is an attention based model that provides feature importance over time by learning dual attention scores (over time and features).

4. Others: In this class of methods we have two baselines: (a) **LIME** [25]: One of the most common explainabilty methods that assigns local importance to input features. Although LIME isn't designed to assign temporal importance, as a baseline, we use LIME at every time point for a fair comparison. (b) **Gradient-SHAP** [21][1]: This is a gradient-based method to compute Shapley values used to assign instance-level importance to features and is a common baseline for static data. Similar to LIME, we evaluate this baseline at every time-point.

## 4.1    Simulated Data

Evaluating the quality of explanations is challenging due to the lack of a gold standard/ground truth for the explanations. Additionally, explanations are reflective of model behavior; therefore, such evaluations are tightly linked to the reliability of the model itself. To account for that, we evaluate the functionality and performance of our baselines in a controlled simulated environment where the ground truth for feature importance is known. A description of all datasets used is presented below:

**Spike Data:**    We simulate a time series data where the outcome (label) changes to 1 as soon as a spike is observed in the relevant feature (and is 0 otherwise). We keep this task fairly simple to ensure that the black-box classifier can indeed learn the right relation between the important feature and the outcome, which allows us to focus on evaluating the quality of the importance assignment. We expect the explanations to assign importance to only the one relevant feature, at the exact time of spike, even in the presence of spikes in other non-relevant features. We generate $D = 3$ (independent) sequences of standard non–linear auto-regressive moving average (NARMA) time series. We add linear trends to the features and introduce random spikes over time for every feature. We train an RNN-based black-box predictor with AUC$= 0.99$, and choose feature 0 to be the important feature that determines the output. The full procedure is described in Appendix B.2.

**State Data:**    The first simulation ensures the correct functionality of the method but does not necessarily evaluate it under complex state dynamics that are common in real-word time series data. The state data has more complex temporal dynamics, consisting of multivariate time series signals with 3 features. We use a non–stationary Hidden Markov Model with 2 latent states and a linear transition matrix to generate observations over time. Specifically, at each time step $t$, observations are sampled from the multivariate normal distribution determined by the latent state of the HMM. The state transition probabilities are modeled as a function of time to induce non-stationarity. The outcome $y$ is a Bernoulli random variable, which, in state 1, is only determined by feature 1, and in

state 2, by feature 2. The ground truth importance for observation $x_{i,t}$ is 1 if $i$ is the important feature and $t$ is the time of state change.

**Switch-Feature Data:** This dataset is a more complex version of state data where features are sampled according to a Gaussian Process (GP) mixture model. Due to the shift in GP parameters at state-transitions, a shift is induced in the predictive distribution, and the feature that drives the change in the predictive distribution should be appropriately scored higher at the state-transitions. Thus, here, the quality of the generator used to characterize temporal dynamics reliably is critical.

| Method | AUROC (explanation) | AUPRC (explanation) | AUROC drop (black-box) | Acc. drop (black-box) |
|---|---|---|---|---|
| FIT | **0.968±0.020** | 0.866±0.022 | 0.323±0.011 | 0.113±0.007 |
| AFO | 0.942±0.002 | **0.932±0.006** | 0.344±0.012 | **0.118±0.007** |
| FO | 0.943±0.001 | 0.894±0.026 | 0.231±0.016 | 0.112±0.070 |
| Deep-Lift | 0.941±0.002 | 0.520±0.098 | 0.356±0.131 | 0.109±0.065 |
| IG | 0.926±0.006 | 0.671±0.030 | **0.375±0.146** | 0.099±0.058 |
| RETAIN[2] | 0.249±0.168 | 0.001±0.000 | 0.001±0.007 | -0.002±0.006 |
| LIME | 0.926±0.003 | 0.010±0.001 | 0.003±0.002 | 0.008±0.006 |
| GradSHAP | 0.933±0.004 | 0.516±0.039 | 0.295±0.105 | 0.088±0.049 |

Table 1: Performance report on Spike data.

**Results:** We evaluate our method using the following metrics[3]:

1. Ground-truth test: We evaluate the quality of feature importance assignment across baselines compared to ground-truth using AUROC and AUPRC.

2. Performance Deterioration test: Additionally, we perform a Performance Deterioration Test to measure the drop in the predictive performance of the black-box model when the most important observations are omitted. Note that all baseline methods provide an importance score for every sample at each time point. Assuming that an important observation is more informative, removing it from the data should worsen the overall predictive performance of the black-box model. Hence a larger performance drop indicates better importance assignment. Since the data is in the form of time series and observations are correlated in time, we cannot eliminate the effect of an observation simply by removing that single measurement. Therefore, we instead carry-forward the previous measurement.

| Method | State Data | | | Switch-Feature data | | |
|---|---|---|---|---|---|---|
| | AUROC (explanation) | AUPRC (explanation) | AUROC drop (black-box) | AUROC (explanation) | AUPRC (explanation) | AUROC drop (black-box) |
| FIT | **0.798±0.027** | **0.171±0.017** | **0.71±0.036** | **0.720±0.013** | **0.159±0.005** | **0.46±0.011** |
| AFO | 0.554±0.002 | 0.025±0.000 | 0.504±0.005 | 0.590±0.001 | 0.048±0.001 | 0.200±0.018 |
| FO | 0.538±0.002 | 0.023±0.000 | 0.264±0.023 | 0.509±0.002 | 0.031±0.000 | 0.058±0.012 |
| Deep Lift | 0.550±0.006 | 0.038±0.001 | 0.047±0.009 | 0.523±0.006 | 0.043±0.001 | 0.044±0.008 |
| IG | 0.565± 0.002 | 0.041± 0.001 | 0.043±0.001 | 0.545±0.002 | 0.045±0.002 | 0.127 ±0.031 |
| RETAIN | 0.510±0.017 | 0.031±0.003 | 0.044±0.026 | 0.521±0.005 | 0.034±0.000 | 0.010±0.005 |
| LIME | 0.482±0.004 | 0.027±0.000 | -0.040±0.028 | 0.529±0.004 | 0.034±0.001 | 0.071±0.053 |
| GradSHAP | 0.486±0.002 | 0.024±0.000 | 0.252±0.017 | 0.506±0.002 | 0.036±0.001 | 0.143±0.003 |

Table 2: Performance report on State and Switch feature data

We restrict our evaluation to subsets of size one for a fair comparison with baselines. Tables (1) and (2) summarize performance results from all simulation settings. For a simple task like spike data, almost all baselines assign correct importance to the relevant spike. However, due to the existing

linear trend in our autoregressive features, perturbation methods are noisy as they sample unlikely or out of domain counterfactuals (see Appendix B for visualization of different methods). By ignoring shifts in predictive distributions, gradient methods in-turn pick up on all spikes in the relevant feature whereas FIT, and perturbation methods correctly pick up only the first spike. LIME misses the spike as they are rare and therefore suffers from a low AUPR as well as low-performance drop.

In both state experiments, FIT outperforms all baselines since it is the only method that assigns importance to correct time points. Attention and Gradient-based methods can pick up the important feature in each state, but persist importance within the state, meaning that they fail to identify the important time. Perturbation based methods also perform very similarly to gradient-based methods; however, the randomness in the perturbations results in noisier importance assignments.

## 4.2 Clinical Data

| Method | MIMIC-mortality | | MIMIC-intervention | |
| --- | --- | --- | --- | --- |
| | AUROC drop $(95 - pc)$ | AUROC drop $(k = 50)$ | AUROC drop $(95 - pc)$ | AUROC drop $(k = 50)$ |
| FIT | **0.046±0.002** | **0.133±0.025** | 0.024±0.004 | 0.050±0.012 |
| AFO | 0.023±0.003 | 0.068±0.003 | -0.00±0.00 | 0.025±0.005 |
| FO | 0.028±0.006 | 0.095±0.042 | 0.022±0.005 | 0.058±0.009 |
| Deep-Lift | 0.045±0.004 | 0.067±0.038 | 0.024±0.005 | 0.066±0.015 |
| IG. | 0.036±0.003 | 0.056±0.014 | 0.023±0.005 | 0.068±0.015 |
| RETAIN | 0.020±0.014 | 0.032±0.019 | NA | NA |
| LIME | 0.028±0.000 | 0.087±0.000 | **0.028±0.008** | 0.064±0.018 |
| GradSHAP | 0.036±0.000 | 0.065±0.062 | 0.025±0.005 | **0.072±0.017** |

Table 3: Performance report on MIMIC-mortality and MIMIC-intervention data.[4]

Explaining models based on feature and time importance is critical for clinical settings. Therefore, having verified performance quantitatively on simulated data, we test our methods on a more complex clinical data set, called MIMIC III, that consists of de-identified EHRs for $\sim 40,000$ ICU patients at the Beth Israel Deaconess Medical Center, Boston, MA [17]. We use time series measurements such as vital signals and lab results for 2 evaluation tasks, with 2 distinct black-box models respectively.

1. MIMIC Mortality Prediction: An RNN-based prediction model that uses static patient information (age, gender, ethnicity), 8 vital and 20 lab measurements to predict mortality in the next 48 hours.

2. MIMIC Intervention Prediction Model: This model predicts a set of non-invasive and invasive ventilation and vasopressors using the same set of features as above. Further details of both black-box models and data can be found in Appendix B.5.

**Results:** Due to the lack of ground-truth explanation for real data, we use a number of performance deterioration tests to evaluate the quality of explanations generated by different baseline methods. Specifically, we use the following performance deterioration tests: i) 95-percentile test: drop observations with an importance score in the top 95-percentile of the score distribution according to each baseline. ii) $k$-drop test: remove the top $k$ most important observations for each time series sample. Note that for $k$-drop test, we have selected patients who undergo significant change in health state – i.e. patients with highest change in risk of mortality $(> 0.85)$ or a mean likelihood of intervention status $(> 0.25)$ in the 48-hours of ICU stay. This ensures that the top observations that are omitted are significant scores.

## 5 Related Work

Different classes of approaches have been proposed for explaining and understanding the behavior of time series models. A number of these approaches, like attention mechanisms, are explicitly designed

for time series settings, while others are originally designed for static supervised learning tasks. In *parameter visualization*, recurrent model behavior is explained via visualization of latent activations of deep neural networks [30, 27, 23, 20]. This approach helps experts understand or debug a model, but it is too refined to be useful to the end-users like clinicians due to the complexity of the network and latent representations.

While *Attention Mechanism* was originally developed for machine translation [4, 33, 22], it can also be used to gain insight into time series model behavior. Attention models are suitable for sequential data and boost performance by learning to reliably learn from long range dependencies [29, 18, 34, 8]. The parameters in these models, called attention weights, are used to explain model behavior in time. However, due to the complex mappings to latent space in recurrent models, attention weights cannot be directly attributed to individual observations of a time series [13] To address this issue, Choi et al. [8] propose to augment the attention mechanism using separate sets of parameters to obtain importance scores over time and features, while [14] modifies hidden states of an LSTM in combination with a mixture attention framework to get variable importance.

*Attribution methods* explain models by evaluating the importance of each feature on the output [28, 37, 26, 31], also commonly known as Saliency methods in the image domain [1, 24]. Attributions can be assigned using gradient-based methods to assess the sensitivity of the output to small changes in the input features [3, 35, 15, 31, 26]. Other methods locally approximate classifiers linearly in order to explain model outcome with respect to a local reference [25, 21, 19]. Most of these methods have been designed for static data like images and need to be carefully augmented for time series settings since the vanishing gradients in recurrent structures compromise the quality of feature importance assignment [16]. L2X [7] and INVASE [36] explicitly design explainers that select a subset of relevant features approximating the black-box model reliably using mutual information and KL-divergence, respectively.

*Perturbation methods* assign importance to a feature based on changes to model output by perturbing the features [37, 38]. This is often approximated by replacing the feature with mean value or random uniform noise [11, 10]. A criticism of such methods is that the noisy perturbations can be out-of-domain for specific individuals and can lead to explanations that are not reflective of the systematic behavior of the model [6]. As a more stable and reliable alternative, Burns et al. [5] frame black box model interpretability as a multiple hypothesis testing problems, where they assess the significance of the change in model outcome as a result of replacing features with counterfactuals [12, 5] .

# 6 Discussion

This work proposes a general approach for assigning importance to observations in a multivariate time series, based on the amount of information each observation adds toward approximating the output distribution. We compare the proposed definition and algorithm to several existing approaches and show that our method is superior at localizing important observations over time by explicitly modeling the temporal dynamics of the time series data. FIT is model agnostic and can be used to assign importance scores over a subset of observations for arbitrarily complex models and different time series data. Since FIT can provide real time explanations for any subset of observations, we see the potential for it to be used for real-time data acquisition along with providing feature importance scores. A future line of work can investigate the ability of our score to find optimal subsets to acquire. Also, although out of scope for this work, our method is extendable to non-instantaneous attributions. This requires 1) evaluating temporal shift (with appropriate delays, e.g., by binning epochs over time); 2) having a conditional generator that models distribution over multiple time-steps. Such modifications to the generator are also useful when *gradual shifts* like trends occur in the data. For explanations of models used for longer term disease management, like chronic conditions, we would suggest using multi-step predictions.

## Acknowledgments and Disclosure of Funding

Resources used in preparing this research were provided, in part, by the Province of Ontario, the Government of Canada through CIFAR, and companies sponsoring the Vector Institute. This research was undertaken, in part, thanks to funding from the Canada Research Chairs Program, the Canadian Institute of Health Research (CIHR), the Natural Sciences and Engineering Research Council of Canada (NSERC), and the DARPA Explainable AI (XAI) Program.

## Broader Impact

This work adds to the growing body of literature in instance-level feature importance attribution and lies in the general purview of explainable machine learning. We focus on the time-series domain, where not many methods have been developed. We are primarily motivated by potential applicability to clinical settings, where time-series data is widely collected in real-time and in Electronic Health Records or EHRs. This preliminary proof-of-concept serves to determine the utility of the proposed framework and highlights specific benefits compared to the existing literature in this area.

While there is no clear consensus on the practical utility of explainable ML and its implication on the trust and safety of end-users, we believe there is a general benefit to grounding this class of methods methodologically. Our work is a step toward that. We additionally evaluate the method with preliminary reliability tests. We believe this is a crucial first step toward improving the framework for practical deployment. At this juncture, the proposed method's application is currently unsafe without further testing on the deployment data and further understanding the nature of the underlying clinical tasks where such a method may be used.

## Footnotes

[1]https://captum.ai/docs/algorithms#gradient-shap

[3]https://github.com/sanatonek/time_series_explainability/tree/master/TSX

[4]Since MIMIC-Intervention is a multi-label prediction task, RETAIN could not be reproduced on this task.

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
