[Supplementary Material]

# A Appendix

## A.1 Notation

Summary of notations used throughout the paper

| Notation | Description |
|---|---|
| $[K]$ for integer $K$ | Set of indices $[K] = \{1, 2, \ldots, K\}$. |
| $d, t$ | Index for feature $d$ in $[D]$, time step $t$ |
| $-d$ | Set $\{1, 2, \cdots, D\} \setminus d$ |
| $\mathbf{S} \subseteq \{1, 2, \cdots, D\}$ | Subset of observations |
| $\mathbf{S}^c$ | Set $\{1, 2, \cdots, D\} \setminus \mathbf{S}$ |
| **Data** | |
| $x_{i,t}$ | Observation $i$ at time $t$. |
| $\mathbf{x}_{\mathbf{S},t}$ | Subset of observation $\mathbf{S}$ at time $t$. |
| $\mathbf{x}_t \in \mathbb{R}^d$ | Vector $[x_{1,t}, x_{2,t}, \cdots, x_{d,t}]$ |
| $\mathbf{X}_{0:t} \in \mathbb{R}^{d \times t}$ | Matrix $[\mathbf{x}_0; \mathbf{x}_1; \cdots; \mathbf{x}_t]$ |
| $p(y_t|\mathbf{X}_{0:t}) \triangleq f(\mathbf{X}_{0:t})$ | Outcome of the model $f$, at time $t$ |

Table 4: Notation used in the paper.

## A.2 Toy example explaining FIT scores

Consider a setup with $D = 2$ features, and the true outcome random variable $y_t = 2x_{1,t} + 0x_{2,t} + \epsilon_t \forall t \in \{1, 2, \cdots, T\}$, where $\epsilon$ is a noise variable independent of $\mathbf{x}_t$ (no auto-regression). Assume that all features are independent. Let $x_{1,t=0} = 0$ and $x_{1,t=1} = 1$. Finally let the distribution shift from time-step 0 to 1 be $\mathrm{KL}(p(y|\mathbf{X}_{0:1}) \parallel p(y|\mathbf{x}_0)) = C$. Consider the setup of figuring out the best observation to acquire at time step 1. The first term (T1) for all singleton sets is fixed and equal to $C$. Since observing $x_2$ has no effect on the outcome, T2=T1 or $\mathrm{KL}(p(y|\mathbf{X}_{0:1}) \parallel p(y|\mathbf{x}_0, x_{2,t})) = \mathrm{KL}(p(y|\mathbf{X}_{0:1}) \parallel p(y|\mathbf{x}_0))$ and the score $I(x_2, t) = 0$. Now consider feature 1. Since observing $x_{1,t=1}$ is sufficient to predict $y$ at time $t = 1$, T2 in this case $\mathrm{KL}(p(y|\mathbf{X}_{0:1}) \parallel p(y|\mathbf{x}_1)) = 0$ and $I(x_1, t) = C$. That is, $\{1\}$ completely explains the distributional shift. This example demonstrates the following compelling properties of the score.

## A.3 Generative Model for Conditional Distribution

We approximate the conditional distribution using a recurrent latent variable generator model $\mathcal{G}$, as introduced in [9]. The latent variable $Z_t$ is the representation of the history of the time series up to time $t$, modeled with a multivariate Gaussian with a diagonal covariance. The conditional distribution of $\mathbf{x}_t$ is modeled as a multivariate Gaussian with full covariance, using the latent sample $Z_t$.

Figure 2: Graphical model representation of the conditional generator. $Z_t$ is the latent representation of the signal history up to time $t$. The counterfactual $\hat{\mathbf{x}}_{t+1}$ will be sampled from the distribution generated by the latent representation

# B Simulated Data

## B.1 Spike Data

To simulate these data, we generate $D = 3$ (independent) sequences as a standard non–linear auto-regressive moving average (NARMA) time series. Note that we also add linear trends to features 1

and 2 of the form:

$$x(t+1) = 0.5x(t) + 0.5x(t)\sum_{i=0}^{l-1} x(t-l) + 1.5u(t-(l-1))u(t) + 0.5 + \alpha_d t \qquad (4)$$

for $t \in [80]$, $\alpha > 0$ (0.065 for feature 2 and 0.003 for feature 1), and order $l = 2$, $u \sim \mathcal{N}(0, 0.03)$. We add spikes to each sample (uniformly at random over time) and for every feature $d$ following the procedure below:

$$y_d \sim \text{Bernoulli}(0.5);$$

$$\eta_d = \begin{cases} \text{Poisson}(\lambda = 2) & \text{if } \mathbf{1}(y_d == 1) \\ 0 & \text{otherwise} \end{cases} \qquad (5)$$

$$\mathbf{g}_d \sim \text{Sample}([T], \eta_d); \ x_{d,t} = x_{d,t} + \kappa \, \forall t \in \mathbf{g}_d$$

where $\kappa > 0$ indicates the additive spike. The label $y_t = 1 \, \forall t > t_1$, where $t_1 = \min g_d$, i.e. the label changes to 1 when a spike is encountered in the first feature and is 0 otherwise. We sample our time series using the python TimeSynth[5] package.

FIT generator trained for this data $\mathcal{G}_i$ is a single layer RNN (GRU) with encoding size 50. The total number of samples used is 10000 (80-20% split) and we use Adam optimizer for training on 250 epochs. Additional sample results for the Spike experiment are provided in Figures 3 for an RNN-based prediction model. Each panel in the figure shows importance assignment results for a baseline method.

## B.2  State Data

In this dataset, the random states of the time series are generated using a two state HMM with $\pi = [0.5, 0.5]$ and transition probability $T$:

$$T = \begin{bmatrix} 0.1 & 0.9 \\ 0.1 & 0.9 \end{bmatrix}$$

The time series data points are sampled from the distribution emitted by the HMM. The emission probability in each state is a multivariate Gaussian: $\mathcal{N}(\mu_1, \Sigma_1)$ and $\mathcal{N}(\mu_2, \Sigma_2)$ where $\mu_1 = [0.1, 1.6, 0.5]$ and $\mu_2 = [-0.1, -0.4, -1.5]$. Marginal variance for all features in each state is 0.8 with only features 1 and 2 being correlated ($\Sigma_{12} = \Sigma_{21} = 0.01$) in state 1 and only 0 and 2 on state 2 ($\Sigma_{02} = \Sigma_{20} = 0.01$).

The output $y_t$ at every step is assigned using the $logit$ in 6. Depending on the hidden state at time $t$, only one of the features contribute to the output and is deemed influential to the output. In state 1, the label $y$ only depends on feature 1 and in state 2, label depends only on feature 2.

$$p_t = \begin{cases} \frac{1}{1+e^{-x_{1,t}}} & s_t = 0 \\ \frac{1}{1+e^{-x_{2,t}}} & s_t = 1 \end{cases} \qquad (6)$$

$$y_t \sim Bernoulli(p_t)$$

Our generator ($\mathcal{G}_i$) is trained using a one layer, forward RNN (GRU) with encoding size 10. The generator is trained using the Adam optimizer over 800 time series sample of length 200, for 100 epochs. Additional examples for state data experiment are provided in Figure 4.

## B.3  Switch-Feature Data

In this dataset, the random states of the time series are generated using a two state HMM with $\pi = [\frac{1}{3}, \frac{1}{3}, \frac{1}{3}]$ and transition probability $T$:

$$T = \begin{bmatrix} 0.95 & 0.02 & 0.03 \\ 0.02 & 0.95 & 0.03 \\ 0.03 & 0.02 & 0.95 \end{bmatrix}$$

Figure 3: Additional examples from the Spike data experiment

Figure 4: Additional examples from the state data experiment

The time series data points are sampled from the distribution emitted by the HMM. The emission probability in each state is a Gaussian Process mixture with means $\mu_1 = [0.8, -0.5, -0.2]$, $\mu_2 = [0, -1.0, 0]$, $\mu_3 = [-0.2, -0.2, 0.8]$. Marginal variance for all features in each state is $0.1$. The Gaussian Process mixture over time is governed by an RBF kernel with $\gamma = 0.2$.

The output $y_t$ at every step is assigned using the $logit$ in 7. Depending on the hidden state at time $t$, only one of the features contribute to the output and is deemed influential to the output. In state 1, the label $y$ only depends on feature 1 and in state 2, label depends only on feature 2.

$$p_t = \begin{cases} \frac{1}{1+e^{-x_{1,t}}} & s_t = 0 \\ \frac{1}{1+e^{-x_{2,t}}} & s_t = 1 \\ \frac{1}{1+e^{-x_{3,t}}} & s_t = 2 \end{cases} \quad (7)$$

$$y_t \sim Bernoulli(p_t)$$

The generator structure is similar to the one used in the State dataset. Additional examples for state data experiment are provided in Figure 5.

Figure 5: Additional examples from the State data experiment

## B.4 Generator Quality

Figure 6: Conditional generator likelihood loss during training

| Generator | AUROC | AUPRC |
|---|---|---|
| Conditional | **0.72±0.01** | **0.15±0.00** |
| Carry-forward | 0.53±0.00 | 0.03±0.00 |
| Mean Imp | 0.48±0.004 | 0.03±0.00 |

Table 5: Explanation performance of FIT using different generator models.

We compare the performance of our generator with simpler approaches for approximating the conditional, such as carry-forward or mean imputation (Table 5). FIT is flexible to the choice of any generator, however, modelling proper conditional distribution is important when time-series data shows significant shifts where carry-forward and mean imputation will result in noisy scores. To demonstrate the quality of the conditional generator, we have also added the likelihood plots, which show that the generator is not overfitting.

## B.5 MIMIC-III Data

### B.5.1 Feature selection and data processing:

For this experiment, we select adult ICU admission data from the MIMIC-III dataset. We use static patients' information (age, sex, etc.), vital measurements and lab result for the analysis. Table 6 presents a full list of clinical measurements used in this experiment.

**MIMIC-III Mortality Prediction:**   The task in this experiment is to predict 48 hour mortality based on 48 hours of clinical data. The predictor model takes in new measurements every hour, and updates the mortality risk. We quantize the time series data to hour blocks by averaging existing measurements within each hour block. We use 2 approaches for imputing missing values: 1) Mean imputation for vital signals using the sklearn SimpleImputer [6], 2) forward imputation for lab results, where we keep the value of the last lab measurement until a new value is evaluated. We also removed patients who had all 48 quantized measurements missing for a specific feature. Overall, 22,988 ICU admissions were extracted and training process was on a 65%,15%,20% train, validation, test set respectively.

**MIMIC-III Intervention Prediction:**   The predictor black-box model in this experiment is a multilabel prediction that takes new measurements every hour and updates the likelihood of the patient being on Non-invasive, Invasive ventilation, Vasopressor and Other intervention. All features are processed as described above.

### B.5.2 Implementation details:

The mortality predictor model is a recurrent network with GRU cells. All features are scaled to $0$ mean, unit variance and the target is a probability score ranging $[0, 1]$. The model achieves $0.7939 \pm 0.007$ AUC on test set classification task. Detailed specification of the model are presented in Table 7. The conditional generator is a recurrent network with specifications shown in 8.

| Data class | Name |
|---|---|
| Static measurements | Age, Gender, Ethnicity, first time admitted to the ICU? |
| Lab measurements | LACTATE, MAGNESIUM, PHOSPHATE, PLATELET, POTASSIUM, PTT, INR, PT, SODIUM, BUN, WBC |
| Vital measurements | HeartRate, DiasBP, SysBP, MeanBP, RespRate, SpO2, Glucose, Temp |

Table 6: List of clinical features for the risk predictor model

| Setting | value (MIMIC-III Mortality) | value (MIMIC-III Intervention) |
|---|---|---|
| epochs | 80 | 30 |
| Model | GRU | LSTM (2 layers) |
| batch size | 100 | 256 |
| Encoding size ($m$) | 150 | 128 |
| Loss | Cross Entropy | Multilabel Binary Cross entropy |
| Regressor Activation | Sigmoid | Sigmoid (4 heads) |
| Batch Normalization | True | True |
| Dropout | True (0.5) | True (0.4) |
| Gradient Algorithm | Adam (lr $= 0.001$, $\beta_1 = 0.9$, $\beta_2 = 0.999$, weight decay $= 0$) | Adam (lr $= 0.001$, $\beta_1 = 0.9$, $\beta_2 = 0.999$, weight decay $= 1e-4$) |

Table 7: Mortality risk predictor model features.

| Setting | value |
|---|---|
| epochs | 150 |
| RNN cell | GRU |
| batch normalization | True |
| batch size | 100 |
| RNN encoding size | 80 |
| Regressor encoding size | 300 |
| Loss | Negative Log-likelihood |
| Gradient Algorithm | Adam (lr $= 0.0001$, $\beta_1 = 0.9$, $\beta_2 = 0.999$, weight decay $= 0$) |

Table 8: Training Settings for Feature Generators for MIMIC-III Data (Mortality and Intervention task)

| Method | State data (sec) $t = 100, d = 3$ | Switch feature data (sec) $t = 100, d = 3$ | MIMIC data (sec) $t = 48, d = 27$ |
|---|---|---|---|
| FIT | 101.05 | 101.16 | 352.65 |
| AFO | 75.614 | 75.4181 | 190.448 |
| Deep Lift | 12.551 | 12.9523 | 5.056 |
| Integrated Grad. | 295.44 | 297.205 | 126.161 |
| RETAIN | 0.2509 | 0.2426 | 0.4451 |

Table 9: Run-time results for simulated data and MIMIC experiment.

## B.6   Run-time analysis

In this section we compare the run-time across multiple baselines methods on a machine with Quadro 400 GPU and Intel(R) Xeon(R) CPU E5-1620 v4 @ 3.50GHz CPU. The results are reported in Table 9, and represent the time required for evaluating importance value of all feature over every time step for a batch of samples of size 200.

## B.7   Subset Importance

Assigning importance to a subset of features is a novel property of FIT. To provide results on this, we identified subsets of correlated features using hierarchical clustering on Spearman correlations for MIMIC-III (mortality prediction task) and used FIT to evaluate the scores assigned to these subsets. Results for this analysis are provided in Table 10.

| Subset | AUROC drop |
|---|---|
| S1 (['ANION GAP', 'CREATININE', 'LACTATE', 'MAGNESIUM', 'PLATELET', 'SODIUM']) | 0.007±0.000 |
| S2 (['ALBUMIN', 'BILIRUBIN', 'POTASSIUM', 'PTT', 'INR']) | 0.005±0.002 |
| S3 (['HeartRate', 'SysBP', 'DiasBP']) | 0.004±0.003 |
| S4 (['GLUCOSE', 'SpO2']) | 0.004±0.002 |
| S5 (['BICARBONATE', 'CHLORIDE', 'HEMATOCRIT', 'HEMOGLOBIN', 'PHOSPHATE', 'PT', 'BUN', 'WBC', 'MeanBP', 'RespRate', 'Glucose', 'Temp']) | 0.011±0.015 |

Table 10: Subset performance drop on MIMIC

## B.8   Sanity Checks

Figure 7: Deterioration in Spearman rank order correlation between importance assignment of the original model to a randomized model.

We further evaluate the quality of FIT using the parameter randomization test previously proposed as a sanity check for explanations [1]. We use cascading parameter randomization by gradually randomizing model weights. We measure the rank correlation of explanations generated on the randomized model and the explanation of the original model. A method is reliable if its explanations of the randomized model and original model are uncorrelated, with increased randomization further reducing the correlation between explanations. Figure 7 shows that FIT passes this randomization test.

## Footnotes

[5]https://github.com/TimeSynth/TimeSynth

[6] https://scikit-learn.org/stable/modules/generated/sklearn.impute.SimpleImputer.html