[Reviews · NeurIPS 2020]

Review 1

Summary and Contributions: This paper proposes a novel method (FIT) for attributing feature importance in the context of time series data. The method calculates the importance of a set of features by integrating over their complement and comparing the predictive distributions via their KL divergence. Some promising results are provided on synthetic data and on MIMIC-III mortality and intervention models.

Strengths: In this work, the authors extend information theoretic approaches to feature importance to the sequential context. Unlike previous work (e.g. L2X, INVASE), this method incorporates temporal structure, and that in two important ways: (i) taking account of the natural temporal shift of the predictive distribution due to the model dynamics; and (ii) integrating over features *conditioned* on previous values. The method demonstrates strong performance on synthetic supervised switching data as well as on mortality prediction for MIMIC-III (intervention prediction performed less well).

Weaknesses: 1. Focus on instantaneous relationships. The paper focuses exclusively on cases where the predictive distribution shifts instantaneously based on certain features. There are a great many examples -- especially in healthcare (e.g. drugs) -- where the instantaneous shifts are negligible, but the long term effects are substantial. There is no discussion regarding this important shortcoming. The mortality/intervention prediction experiments obviate this problem by defining the target to be 48 hours in the future, but this is not a generally applicable approach. Even if one were to pre-define the target in this way, FIT still cannot attribute appropriately where complex dependencies exist in time. As a simple example of this, consider a model that has learned that the prediction of y_t should change based on a logical AND of (x_{t-1,i} = 1) & (x_{t,i} = 1). The first of these inputs will be assigned no importance by FIT, only the second will obtain an attribution. The former is accounted for in the temporal shift (T1) in eq. (1) which is not involved in attribution. This shortcoming (so far as I understand it) does not apply to competitors such as FO, RETAIN, IG etc. 2. Non-general conditional distribution of features. [L102]: Modelling the conditional distribution using a recurrent LVM with full covariance matrix output (i) limits the inputs to real-valued observations -- binary values are common in healthcare scenarios; (ii) limits the dimensionality of $x_t$ due to the sample complexity and efficiency of estimating a full PSD matrix. How would you handle binary variables or large $D$?

Correctness: Insofar as it goes, I am happy that the claims made in the paper are correct.

Clarity: Yes. Easy to read, clear explanations, good use of diagrams.

Relation to Prior Work: Yes. The authors use a different method to previous time series efforts, and the difference in performance to Attention-based methods is covered by reference to RETAIN. However, I am not especially well read in the interpretability literature.

Reproducibility: Yes

Additional Feedback: Feature importance in the context of time series models is a difficult problem, and I applaud the authors for the work so far. If my analysis of the "instantaneous-only" attribution is correct, it may yet be one-tool-of-many that can be used in practice. I may be willing to improve my score with a suitable response, or even if the issue is discussed adequately in the paper. Minor comments: *[L100]: The use of a temporal model for sampling the $x_{S^c}$ is nice. Did you encounter any problems with this SRNN (e.g. overfitting, and hence low diversity of $x_{S^c}$)? * [L114]: "For ease of interpretation, the score I(x_S, t) is normalized between -1 to 1 using a scaled sigmoid function." This raises a difficult question of how to choose the slope/effective truncation. Given that this is not used in the rest of the manuscript, perhaps change to "may be normalized"? * [Table 1]: There are a number of instances where results appear statistically indistinguishable from each other in the table (not always weighted in your favor!). I know this is common in the ML literature, but can I make a request that all such "joint-winners" be highlighted? * [Table 3, column 1]: Deep-Lift looks to be significantly indistinguishable from FIT -- again, can this be highlighted? * A number of refs to the appendix are incorrect. Appendix A.6 is presumably now part of Appendix B. * RETAIN superscript (2) on p6 has no corresponding footnote. * Figure 2: Typo in "Gradient" (Graditen). * [L465]: Eqn ref should be (6) not (7). * [L473]: Two state -> Three state ------------------------------- I thank the authors for their response. I'm especially encouraged by the proposed discussion of non-instantaneous attribution in the main text. While the suggestions to ameliorate this appear to be sound, they are fairly limited and ad-hoc, but within the context of existing XAI work for TS models, I'm happy to raise my score accordingly. Also, notwithstanding the many other important points addressed in the response, I think Table 2 is a key addition to the paper.


Review 2

Summary and Contributions: In this paper, the authors present an approach for time series explainability which looks to identify which subset of features best explain the current black box prediction. The importance is assessed by comparing the KL divergence between the current prediction conditioned on the covariates (immediate past, current) and the additional information provided by current subspace of features. The authors use Monte Carlo method for estimating the conditional joint w.r.t subspace of interest. Empirical results demonstrate effectiveness of the proposed approach.

Strengths: 1) Authors present an intuitive and relatively non-complicated approach for addressing the time series explainability problem from a feature importance perspective. 2) Authors have taken effort to provide code for their algorithm and all baselines considered also.

Weaknesses: 1) One main limitation of the work is that it expects end user to know a set of features (S) to infer explainable insights. Without knowing such prior knowledge one would have to explore all possible subsets of features to use this approach for multiple feature sets. Some sort of submodular approach or greedy heuristic over the feature space (for example, max min hill climbing for pruning the search space in Bayesian Networks - Tsamardinos et al. 2006 which can mitigate this in practice might be useful as data scientists / reliability engineers who are looking to use explainability tools for time series may not may not want to provide the subset of features before validating the explanation from the approach. 2) Another main issue with this paper is that the evaluation section should be re-written to make the insights more clear. I also suggest that the authors consider validating the insights using a SME (like clinical expert) to improve the evaluation section. 3) Figure 2 has a typo - (Gradient based)

Correctness: Yes

Clarity: Yes

Relation to Prior Work: Yes

Reproducibility: Yes

Additional Feedback: 1) The work definitely needs an alternative title. I would suggest not to use the word "wrong" in your title. 2) Mention Output also in Algorithm 1. I have read the author response and comments made by other reviewers also. I have updated my score for this paper accordingly.


Review 3

Summary and Contributions: This submission introduces Feature Importance in Time (FIT), a framework to determine the importance of single features or feature subsets in time series (TS) for each point in time. Features in this context refer to the dimensions/channels of multivariate TS. The core of the approach is to assess the distributional shift of a black-box model when only a subset of features is used. More precisely, the difference of two KL-divergences defines the proposed FIT importance score: The first KL-div. is computed between the output distribution up to time point t-1 and output dist. at time point t. This models the general shift of the distribution between measurements at t-1 and t. The second KL-div. is computed between a counterfactual output distribution at t-1 (considering only a subset of features) and the output dist. at time point t. Simply put, the FIT score measures a hypothetical distribution shift from t-1 to t through feature selection with respect to the observed distribution shift. The authors show intuitive behavior of the FIT score and an information theoretic interpretation of the score. In the experimental evaluation it is shown, that ground truth features (in a synthetic setup) can be found and that removing identified features results in a larger performance drop compared to other methods (on real clinical data).

Strengths: • Relevance to the NeurIPS community: Considering the increasing number of real-world TS data, feature importance detection in TS is of high relevance for the ML community. Especially in fields that can benefit from explainability such as biology or medicine, this method can be helpful. • Theoretical grounding and soundness: The core idea behind FIT is the hypothesis that (an) important feature(s) are responsible for an observed shift of the output distribution while unimportant ones are not. This is a sensible assumption and slightly reminds me of Granger causality where we ask, does a certain feature add value for my prediction, while here the authors ask whether a certain feature is responsible for the observed distribution shift of my predictions. The authors phrase this question from an elegant probabilistic viewpoint in terms of KL-divergences, which make FIT quite versatile. • The implementation seems quite simple and the FIT score has an intuitive interpretation which allows it to become a widely used method for feature importance detection. • On synthetic data FIT outperforms competitors significantly, and on clinical data the removal of identified important features leads to a comparable performance drop. • Novelty: I think the proposed method is a creative and novel way of approaching a rising challenge in TS analysis. While new theoretical insights are missing, I see great potential in the application in healthcare and other fields.

Weaknesses: • For a method about explainability, I am missing a paragraph on interpretable time series features (similar to what is presented in the appendix but more thorough and with a potential clinical interpretation in the case of the MIMIC experiments). The analysis on real data is limited to the performance deterioration test, a measure of limited utility in practice. You could point out the strengths of your method by visualizing importance values on top of time series and comment on their clinical relevance (i.e., does your method actually find clinically relevant features or does it find new important interactions between features?). You might consider moving the sanity experiment to the appendix and insert such experiments instead.

Correctness: As mentioned above, I believe that the core assumption that motivate the proposed method are sensible and all claims seem to be correct. The authors provide standard deviations and don’t oversell the method.

Clarity: Methodology, experiments, and results are very clearly written and one can easily follow the main aspects of the paper. Figure 1 provides a clear and comprehensible overview over the core hypothesis of the method. Data sets, as well as competitors are well described and results are presented in a digestible manner.

Relation to Prior Work: In general, related work is sufficiently discussed, but a recent publication from ICML is missing: https://proceedings.icml.cc/static/paper_files/icml/2020/4750-Paper.pdf

Reproducibility: Yes

Additional Feedback: • One aspect that is not 100% clear to me is how you implement the conditional probability in line 7 of the pseudo-code coming from line 5? • Oftentimes TS motifs (a longer than 1 subsequence of a TS) are important features. I wonder if you evaluated to which extent your method works if you want to analyze such consecutive signals. =============================================================== Dear authors, thank you for your thorough rebuttal. Given my positive score in the first round of reviewing, I will stick to my current rating.


Review 4

Summary and Contributions: This paper proposes FIT (Feature Importance in Time), a framework that can generate time-specific explanations for black-box multivariate time-series models, which are commonly used in healthcare settings. FIT considers an observation (a subset of features at a particular time point) to be important if it can "approximate the model's predictive distribution well, even in the absence of other features." Quantitatively, it assigns importance score by calculating the observation's contribution to the distributional shift, in terms of KL divergence, between the actual predictive distribution and a counterfactual distribution (setting other features at the same time point as the observation to be unobserved). The paper then compares FIT against other interpretation approaches (perturbation-based, gradient-based, attention-based, LIME, Shapley) on both simulated and real clinical data and showed the proposed approach outperforms the baselines in many cases.

Strengths: The paper is well-written in general and explains the relevant concepts and proposed approach clearly. It is well motivated and identified a gap in past research on the subject of feature importance analysis for time-series models. I also liked the use of intuition and gradual build-up when explaining why the importance score was defined in that particular way. I especially liked Fig. 1 pictorial illustration of how the change in conditional distribution corresponded to the different values of the KL terms in the importance score calculation.

Weaknesses: The experimental section could be improved to clarify/include several important details. Please see comments below.

Correctness: The description of the approach is presented clearly with mathematical definitions of the importance scores using both KL divergence and cross entropy. The author also discussed the desired properties in the limiting cases (L82-L87) which helps the reader better understand the proposed approach. The empirical experiments and evaluations appear to be correct, however there are several areas of the experimental section that can be improved/clarified (see comments below).

Clarity: Overall I think the paper is clearly written and very easy to follow. Notation is precise and follows existing literature.

Relation to Prior Work: The paper mentioned various lines of relevant works, but I think it needs a more explicit discussion on how this paper differs from those previous approaches. For example [5,12] also made use of the counterfactuals, so it would be good to elaborate more on how the proposed approach makes use of the counterfactuals in a similar/different way.

Reproducibility: Yes

Additional Feedback: Dear authors, I really appreciate your enthusiastic and thorough response. Many of the things in the response are important to justify the contribution of this work and its limitations (eg, Table 2 experiments, non-instantaneous attribution), so I do hope you will include these new results/discussions in the main paper, in addition to incorporating the remaining suggestions from all reviewers. Given my already positive review, I am keeping my score rating. =============================================================== *Reproducibility* Code is provided as part of supplementary material and appears to be well-documented. I did not run the code. I encourage the authors to share the code on github if the paper is accepted. *Subset importance* - In Sec 3.2 it was mentioned that "joint importance of feature subsets" could be a key characteristics of FIT, but this was never explored in the experiments. - Besides using singleton sets or pre-defined sets, or formulating it as finding the minimal set, one common way to create feature groupings is to look at pairwise correlations between features. See an example at https://scikit-learn.org/stable/auto_examples/inspection/plot_permutation_importance_multicollinear.html#handling-multicollinear-features *Generator* - FIT relies on having a good generator G to generate the counterfactuals. Have you considered simpler approaches like imputation by carrying-forward, mean/median feature value of training data, or some prespecified normal value? It'd be good to include a brief discussion on the pros and cons of these approaches vs using a separate generator model. - On the empirical evaluation, some quantitative results (perhaps training/validation loss?) on the trained generator is needed to convince the reader that the generator used in FIT is properly trained and can produce reliable counterfactuals. - L112 says the counterfactual distribution is conditioned on S. Could you elaborate on how the VRNN handles that? *Experiments* - For the MIMIC tasks, it would greatly add to the paper if there are qualitative case studies, e.g. reviewing with clinicians if possible, why the important feature identified makes sense, for cases with pos/neg label. - Table 1&2 focused on synthetic data experiments and should be consistent in terms of the evaluation metric. Currently, Table 1 had "accuracy drop" whereas Table 2 doesn't. I suggest focusing on AUROC since the synthetic datasets are probably unbalanced. - Sec 4.2 text does not mention Table 3; should highlight the takeaways for readers. - How many Monte-Carlo samples (the value of $L$ in Alg 1) were used in experiments, and how was the value chosen? This can be better justified by providing the mean importance values as one increases the number of MC samples and showing it stabilizes. - L143 mentions "evaluate on multiclass settings" but I don't believe this was done. The experiments are all binary classifications (single- or multi-task). - L195 says "perturbation methods are noisy" but based on Fig. 4 "afo" subplot, I don't believe that's the case. Could you please clarify? *Discussion & Potential limitations* - In discussion you might want to mention/highlight the fact that these interpretability methods provide us with model explanations rather than causal relations; they help us understand the predictive mechanism of a specific model in mapping features to outcomes, but not the true causal relationship between features and outcomes. - The current formulation of FIT considers observations defined as feature subsets at a particular time point. What if the class label depends on a complex pattern in the feature values? I think an RNN would be able to model that, but it's not just a single or a group of features at _one time point_ that's important. For example, if the label only becomes 1 after we see the second spike, it is less clear which observation we should put the importance on. Perhaps one extension would be to consider sets of time points (i.e. time ranges) in addition to sets of features. This could be mentioned as a limitation/future work since it might be beyond the scope of the current paper. *Presentation & writing suggestions* - "AUPR" vs "AUPRC" should be used consistently, same for "AUROC" and "AUC" - L84-L87: might be better to color T1 and T2 same way in Eqn 1 and Fig 1, I missed the notation on my first read through - L87: please define what C is - L115: is "scaled sigmoid function" referring to tanh? - L157: "for to" -> "to" - L175: "if ... if" -> "if ... and" - Table 1: footnote 2 seems to be missing - L196: "out of domain" -> "out-of-domain" - Table 2: first row last column should be written with 3 decimal places (same as the rest of the table) - Tables 1&2 should specify what the error bars are (95%CI or standard errors) - L215: "Non-invasive" doesn't need to capitalized - Table 3: "95 - pc" reads like "95 minus pc"; also (L224) I think it should be referred to as "above the 95th percentile" rather than "top 95-percentile". Are the top 5% observations to be dropped across all observations in the testing set rather than per example? - L449: the "g" in "g_d" should be bolded - L472: "two-state" -> "three-state" HMM

[Author Response · NeurIPS 2020]

We thank the reviewers for their insightful and positive reviews, finding our work well motivated (R4), addressing a
clear gap in existing research (R4) by proposing a creative/novel (R3) and intuitive (R2, R3, R4) method with great
potential (R3). Along with expanded discussion, we have also addressed minor comments in the updated draft. To
summarize, we have proposed a new method, called FIT, that uses KL-divergence to assign instantaneous feature
importance for time-series observations, accounting for temporal data shift. FIT shows promising results on complex
simulated time-series models as well as two tasks on real healthcare data.

**Subset Selection/Importance** (R2, R4): Assigning importance to a subset of features is a novel property of FIT
and we have added additional experiments and discussions regarding this in our paper. For example, based on R4's
suggestion, we identified subsets of correlated features using hierarchical clustering on Spearman correlations for
MIMIC and used FIT to evaluate the scores assigned to these subsets (Table 1). Of course, existing methods such as
greedy sub-modular search, hill climbing (R2) and modern stochastic search can also be easily used with FIT to find the
optimal subset.

**Generative model quality** (R1, R4): As suggested by R4, we now compare the performance of our generator with
simpler approaches for approximating the conditional, such as carry-forward or mean imputation (Table 2). FIT is
flexible to the choice of any generator, however, modelling proper conditional distribution is important when time-series
data shows significant shifts where carry-forward and mean imputation will result in noisy scores. We have added this
discussion and results to the appendix. To demonstrate the quality of the conditional generator, we have also added the
likelihood plots, which show that the generator is not overfitting (R1).

**Instantaneous attribution** (R1): Based on R1's suggestion, we have added the following to the draft: "Instantaneous
attribution is valuable to understand the additive information of a *new* observation, particularly for real-time predictions.
For example, when managing sepsis in an ICU, instantaneous changes are likely to drive model prediction." We also
highlight how FIT may be extended to non-instantaneous attribution: "Though out of scope for this work, our method
is extendable to non-instantaneous attributions. This requires: 1) evaluating temporal shift (with appropriate delays,
e.g. by binning epochs over time); 2) a conditional generator that models distribution over multiple time-steps. Such
modifications to the generator are also useful when *gradual shifts* like spikes and trends occur in the data (R2, R4). For
explanations of models used for longer term disease management, like chronic conditions, we would suggest using
multi-step predictions." Finally, in the logical AND example (R2), we note that all methods will fail when used for
instantaneous attribution. This is because the score itself from FO and other methods is biased due to issues of vanishing
gradients common in RNNs (Ismail et al. NeurIPS2019). Similarly, no guarantees exist for RETAIN to assign equal
importance to both $x_{t-1,i}$ and $x_{t,i}$.

**Subject matter expert (SME) evaluation** (R2, R3): We asked a clinical collaborator (SME) to annotate important
observations over time and we evaluated FIT scores against these. High positive FIT scores were correlated with
time-points the clinician identified as important in their decision making. Figure 1 shows an example of such annotations
(in red) for 2 different signals from 2 individuals. The clinician determined that patient 1 (top) was tachycardic towards
the end (hour 32) and the FIT scores for Heart rate highlight this time point clearly. As R3 suggested, we have also
added visualizations for MIMIC experiments along with the clinical insight of the SME.

**High-dimensional and binary data** (R1): We agree that high dimensionality of the feature-space can increase sample-
complexity of estimating a full covariance matrix. We have included the following discussion addressing this: "For
high-dimensional data, low-rank approximations can be considered in practice that will reliably model desirable
dependencies efficiently. Binary as well as heterogeneous data-types can be incorporated with recent advances in
heterogeneous data modeling using recurrent models (e.g. Liu et al. AAAI 2018). For more complex data, FIT can use
other conditional generators such as GAIN and Imputation-GANs."

**Expand discussion on insights** (R2, R4): We have significantly expanded discussion for added insights. R4: The
main difference between FIT and other counterfactual methods [5,12] is that we use these counterfactuals to estimate
temporal shift, while [5,12] assess perturbations in model output. Also, the counterfactuals sampled using our generator
marginalize over complement of the target set. Note that such explanations do not provide causal insights but help
understand the predictive mechanism of a model.

| Subset | AUROC drop |
|--------|------------|
| S1 | $0.007 \pm 0.000$ |
| S2 | $0.005 \pm 0.002$ |
| S3 | $0.004 \pm 0.003$ |
| S4 | $0.004 \pm 0.002$ |
| S5 | $0.011 \pm 0.015$ |

Table 1: Subset perf. drop on MIMIC

| Generator | AUROC | AUPRC |
|-----------|-------|-------|
| Conditional | $\mathbf{0.72 \pm 0.01}$ | $\mathbf{0.15 \pm 0.00}$ |
| Carry-forward | $0.53 \pm 0.00$ | $0.03 \pm 0.00$ |
| Mean Imp | $0.48 \pm 0.004$ | $0.03 \pm 0.00$ |

Table 2: Generator quality

Figure 1: Clinical (SME) evaluation

[Meta-Review · NeurIPS 2020]

Three knowledgeable referees support acceptance for the contributions, notably R1, R3 and R4. After the discussion, R2 does not oppose acceptance and I don't see any major criticism to the paper. Therefore, I also recommend acceptance. However, please consider revising your paper to address a common comment by the reviewers regarding non-instantaneous attribution.